# Meta-Inflammation and New Anti-Diabetic Drugs: A New Chance to Knock Down Residual Cardiovascular Risk

**DOI:** 10.3390/ijms24108643

**Published:** 2023-05-12

**Authors:** Alessia d’Aiello, Alice Bonanni, Ramona Vinci, Daniela Pedicino, Anna Severino, Antonio De Vita, Simone Filomia, Mattia Brecciaroli, Giovanna Liuzzo

**Affiliations:** 1Department of Cardiovascular Sciences, Fondazione Policlinico A. Gemelli, IRCCS, 00168 Rome, Italy; alessia.daiello@guest.policlinicogemelli.it (A.d.);; 2Department of Cardiovascular and Pneumological Sciences, Catholic University of Sacred Heart, 00168 Rome, Italy

**Keywords:** type 2 diabetes mellitus, sodium-glucose cotransporter 2 inhibitors, glucagon like peptide 1 receptor agonists, meta-inflammation, cardiovascular diseases

## Abstract

Type 2 diabetes mellitus (DM) represents, with its macro and microvascular complications, one of the most critical healthcare issues for the next decades. Remarkably, in the context of regulatory approval trials, sodium-glucose cotransporter 2 inhibitors (SGLT2i) and glucagon-like peptide 1 receptor agonists (GLP-1 RAs) proved a reduced incidence of major adverse cardiovascular events (MACEs), i.e., cardiovascular death and heart failure (HF) hospitalizations. The cardioprotective abilities of these new anti-diabetic drugs seem to run beyond mere glycemic control, and a growing body of evidence disclosed a wide range of pleiotropic effects. The connection between diabetes and meta-inflammation seems to be the key to understanding how to knock down residual cardiovascular risk, especially in this high-risk population. The aim of this review is to explore the link between meta-inflammation and diabetes, the role of newer glucose-lowering medications in this field, and the possible connection with their unexpected cardiovascular benefits.

## 1. Introduction

Type 2 diabetes mellitus (DM) represents, with its macro and microvascular impairments and its complex pathophysiology, one of the most serious global healthcare issues in forthcoming years [1]. Starting from parental insulin, DM treatment has come a long way, proposing a wide range of oral drugs able to achieve better glycemic control. Interestingly, in the context of regulatory approval trial, sodium-glucose cotransporter 2 inhibitors (SGLT2i) and glucagon-like peptide 1 receptor agonists (GLP-1 RA) showed a reduced incidence of a major adverse cardiovascular event (MACE), namely cardiovascular death and heart failure (HF) hospitalizations [2,3].

Since these findings, growing evidence has been published, trying to understand the pathophysiologic mechanisms underpinning the cardiovascular effects of these new anti-diabetic molecules. Fascinatingly, SGLT2i and GLP-1 RA cardioprotective abilities seem to operate beyond the simplistic glycemic control [4], disclosing a range of pleiotropic effects and a solid link between diabetes and metabolic inflammation. A deeper knowledge of molecular mechanisms sustaining DM as a metabolic and inflammatory disease may help to understand the cardioprotective effects of new anti-diabetic drugs and their role in knocking down residual cardiovascular risk and, above all, the risk of developing HF [5], a heterogeneous clinical syndrome, whose management is ongoing towards major clinical changes [6].

The aim of this review is to explore the link between meta-inflammation and diabetes, the role of newer glucose-lowering medications in this field, and the possible connection with their unexpected cardiovascular benefits.

## 2. Meta-Inflammation, Diabetes and Cardiovascular Disease

The pathogenesis of DM shares with cardiovascular diseases (CAD) a complex landscape of risk factors, including genetic predisposition and various environmental factors like a high-fat diet, sedentary lifestyle, and chronic stress. In particular, CAD is the leading cause of morbidity and mortality in diabetic patients, determining a significant impact on life expectancy. Notably, DM is equivalent to established ischemic CAD risk, and patients with diabetes have a two- to four-fold greater risk of developing CAD than non-diabetic patients [7]. Myocardial infarction, ischemic ictus, and peripheral arterial disease are the main expression of DM progression and, often, the first event in diabetic patients. CAD and DM strictly depend on various inflammatory pathways that are able to promote the onset and development of insulin resistance, atherosclerotic plaque, and HF.

One of the mechanisms linking DM and CAD is the so-called meta-inflammation, i.e., metabolic inflammation, which is a low-grade chronic and sterile inflammatory status maintained by high nutrient intake and which is able to reshape the inflammatory milieu of metabolic cells, tissues, and organs [8]. The starting point of meta-inflammation is the physiological inflammatory trigger prompted by the intake of any meal: in particular, in the postprandial moment, remnants of chylomicrons and very low-density lipoprotein (VLDL, triglyceride-rich lipoproteins) bind to endothelial cells and circulating leukocytes causing acute cellular activation and increased expression of adhesion molecules, cytokines, and oxidative stress characters, such as thiobarbituric acid reactive substances (TBARS), leukocyte O^2−^, and 8-iso-prostaglandin F2alpha (8-PGF2α) [9,10,11]. Triglyceride-rich lipoproteins and free fatty acids (FFAs) raise the expression of vascular cell adhesion protein 1 (VCAM1) in human aortic endothelial cells, favoring leucocyte adhesion [12]. In parallel, an increased neutrophil count coupled with augmented levels of interleukin 6 (IL-6) and hydroperoxides was also observed after high-fat meals in healthy controls [13]. This inflammatory environment promotes insulin resistance by activation of nuclear factor kappa-light-chain-enhancer of activated B cells (NF-κB) and c-Jun N-terminal kinase (JNK) pathways that, in turn, lead to the decline of insulin secretion by pancreatic b-cells [14,15].

This complex system might be the main contributing factor to HF pathophysiology through innate and adaptive system impairment, macrophage functional level shift, accumulation of unfolded proteins [5] within a steady two-way cross-talk low-grade inflammatory state, and metabolic flexibility (Figure 1).

### 2.1. Meta-Inflammation and Metabolic Endotoxemia

Dietary fats can boost levels of circulating bacterial endotoxins, contributing to metabolic endotoxemia, a chronic low-grade elevation of the bacterial component lipopolysaccharide (LPS) with content 10–50-times lower than in septic conditions, which can lead to leukocyte activation, local and systemic inflammation [16]. Recent evidence has proved that metabolic endotoxemia plays a critical role in the inflammatory setting milieu, thereby predisposing to metabolic diseases via pattern recognition receptor engagement, primarily through the toll-like receptor/nod-like receptor (TLR4/NLR) signaling pathway [17,18]. Following a similar pathophysiological framework, both DM and CAD are associated with significant alterations of the gut microbiota, responsible for the increased intestinal permeability and the consequent efflux of LPS into the bloodstream, thus altering the systemic metabolic response [19]. Indeed, already in 1990, it has been demonstrated that patients with end-stage HF show elevated levels of circulating tumor necrosis factor (TNF), which could explain the cachectic condition typical of this syndrome as a consequence of innate immune dysregulation operated by endotoxin-activated monocytes [20,21].

### 2.2. Meta-Inflammation and the “Unfolded Protein Response” (UPR)

The liver and pancreas are the classical target organs in the physiopathology of DM, but adipose tissue is probably at the core of meta-inflammation. Nutrient excess increases adipocyte size up to critical conditions, resulting in reduced vascularization and a hypoxic environment [22], finally precipitating the inflammatory cascade.

In obese mice, adipocytes participate in endoplasmic reticulum stress, a condition accelerated by the accumulation of unfolded/misfolded proteins and able to trigger the protective unfolded protein response (UPR). This process promotes both the synthesis of endoplasmic reticulum-resident chaperone proteins, which encourages protein folding and the protein-degradation mechanism components to achieve new endoplasmic reticulum homeostasis [23]. In addition to these defensive responses, the UPR may also encourage important inflammatory signals, triggering apoptotic and cell death pathways [24]. UPR is significantly enhanced by exposure to nutrient excess, as revealed by experiments in obese and diabetic mice, and its activation promotes inflammation through several mechanisms, i.e., inducing the expression of pro-inflammatory genes by directly acting on the transcription factor activator protein 1 [25]; abnormal nutrient intake pledges the NF-κB signaling pathway [26], promoting the downstream cleavage and activation of the transcription factor cyclic-AMP responsive- element-binding protein H (CREBH), which in turn induces the production of two acute phase proteins, C-reactive protein (CRP) and serum amyloid P-component (SAP) [27]. A similar process has been recorded in HF patients and represents the ground of cardiac hypertrophy, further exacerbated by the acquired immune response.

### 2.3. The Innate Immune Response

An innate immune response is an essential mechanism in triggering meta-inflammation associated with DM and CAD. Pattern recognition receptors can boost meta-inflammation following recognition of both pathogens-associated molecular patterns (PAMPs) and damage-associated molecular patterns (DAMPs). Viral nucleic acids, endotoxins, and peptidoglycans are some of the PAMPs circulating in the bloodstream because of metabolic endotoxemia, while FFAs and self-nucleoproteins are some of the endogenous ligands starring as DAMPs [28]. Pattern recognition receptor activation represents the main sensor to start the inflammation cascade via NF-𝜅B and some interferon regulatory factors (IRFs), which in return trigger pro-inflammatory cytokines and type-1 interferons signaling cascade [29]. These mechanisms could lead to insulin resistance and precipitate in DM via the mediation of macrophage polarization towards the M1 phenotype [30].

In HF, adverse left ventricular (LV) remodeling and LV dysfunction are structural progressions that parallel the activity of innate immunity actors and mediators such as monocyte chemoattractant protein 1 (MCP-1) and progressive monocyte-derived macrophage generation [31].

The innate immune response could be dysregulated at multiple levels, becoming the basement of a meta-inflammatory environment that, in turn, involves multiple cellular compartments.

### 2.4. The Adaptive Immune Response

As far as adaptive immunity is concerned, most evidence proves that CD4^+^T-helper (Th) cells are crucial elements in propagating meta-inflammation [32,33]. The response started by pattern recognition receptors is transferred to T cells by the antigen-presenting cell (APC) and T cell cooperation, which determines the enrolment of these activated cells into the pancreas, adipose tissue, and other target organs, consolidating the meta-inflammation mounted by the native response. The well-known Th polarization, which shapes the adaptive arm of the immune response, is also maintained in DM patients. In particular, in the murine diet-induced obesity model, Kitschier et al. demonstrated early recruitment of Th1 cells into adipose tissue, paving the way for macrophage infiltration and insulin resistance [34]. Notably, several studies on serum cytokine profiling in DM patients reported robust Th1 polarization during the transition from DM to macrovascular complications, especially atherosclerotic coronaropathy, highlighting the role of Th1 polarization in the physiopathology of the disease [35]. Furthermore, a significant action in interfering with insulin signaling and insulin-stimulated glucose uptake is pursued by interferon-gamma (IFN-𝛾), eventually leading to insulin resistance and DM [36,37,38]. Polarized Th1 and Th17 T-cells also play a key role in cardiac fibrosis and adverse cardiac remodeling, sustaining and amplifying the local chronic inflammation leading to HF. In vivo, experimental models demonstrated that CD4^+^ T-cells were expanded in the failing heart, with a Th1/Th2 ratio significantly decreased, whereas the Th17/Treg ratio was increased, underling the loss of anti-inflammatory properties and the gain of the pro-inflammatory counterpart [39].

### 2.5. The Bow Tie Model

Metabolic and inflammatory pathways can converge at many levels, including cell-surface receptors, intracellular chaperones, or nuclear receptors. This molecular army allows robust cooperation between the nutrient-sensing pathways and the immune response aimed at maintaining homeostasis in opposite metabolic and immune circumstances. Unfortunately, this molecular rendezvous may be one of the crucial moments in DM and CAD pathophysiology. Meta-inflammation may be considered the unpredicted consequence of the evolution-driven degeneracy of damage sensors, recently described as a ‘bow tie’ architecture [40]. The main feature of bow tie architecture is the possibility to converge a vast range of inputs (fan in) on an evolutionarily reduced core of components (core), able to translate the inputs into a broad spectrum of outputs (fan out) (Figure 2). Interestingly, the inputs are represented by an extensive range of self and non-self-stimuli, i.e., free fatty acid, LPS, able to bind a restricted number of evolutionarily conserved innate immunity sensors (the bow tie core), whose activation triggers a large number of inflammatory elements [41]. A fascinating example of this promiscuity in immune response receptors is the capacity of saturated fatty acids to turn on both TLR2 and TLR4, important innate immune response receptors involved in pathogen recognition, and trigger the release of pro-inflammatory mediators [42].

This bidirectional cross-talk between metabolic alterations and immune dysregulation is also emerging as a critical component of the pathogenesis of HF, especially for the one with preserved ejection fraction (HfpEF). Indeed, all the alterations mentioned above and driven by metabolic alterations, such as macrophage polarization, accumulation of misfolded proteins, and metabolic reprogramming, contribute to structural and functional remodeling determining HF [5].

The exploration of the meta-inflammation route as a common denominator for DM and CAD might expand the knowledge of molecular pathways underpinning its pathophysiology, thus clearing the unpredicted effects of new anti-diabetic drugs on cardiovascular outcomes.

## 3. Newer Glucose-Lowering Medications and Cardiovascular Effects

### 3.1. Sodium–Glucose Cotransporter 2 Inhibitors (SGLT2i)

The kidneys play a critical role in glucose homeostasis through gluconeogenesis, glucose utilization, and glucose reabsorption. In diabetic patients, renal gluconeogenesis is significantly enhanced, and the capacity to reabsorb glucose in the convoluted segment of the proximal tubule is pathologically increased through the upregulation of SGLT2 transporters [4]. In this context, the SGLT2i have been developed to inhibit the high-capacity, low-affinity SGLT2 receptors in the proximal tubule of the nephron. Interestingly, SGLT2 receptor performance is enhanced in chronic hyperglycemia, determining significant resorption of glucose and sodium while it is downregulated in lower glucose conditions, minimizing the risk of hypoglycemia during SGLT2i therapy [43], notably at lower levels of glycemia or at lower glomerular filtration rate (GFR).

Since their first approval in 2013, SGLT2i have gained a lot of success in the management of diabetic patients, improving glucose control without increasing the risk of hypoglycemia and promoting weight loss with their glucosuric effect. Several clinical trials assessed the safety and efficacy of SGLT2i in patients with diabetes and a large variety of cardiovascular and renal complications. In particular, the EMPA-REG OUTCOME trial proved that diabetic patients at high risk of CAD and treated with empagliflozin had a significant reduction in cardiovascular death and hospitalization for HF [44]. These results have been further assessed for other SGLT2i in large trials, i.e., canagliflozin in CANVAS [45] and CREDENCE [46] and dapagliflozin in DECLARE-TIMI 58 [47].

Probably the most exciting evidence is provided by the DAPA-HF trial, which demonstrated a marked reduction in worsening HF or cardiovascular death on top of HF standard-of-care therapy, with similar effects in patients with and without DM [48] (Table 1).

#### 3.1.1. Possible Targets of SGLT2i Pharmacodynamics

In the last years, several experimental findings suggested a large variety of mechanisms to explain the impressive results of SGLT2i in cardiovascular outcomes. On the one hand, some authors suggested that blood pressure lowering, a “side” effect of SGLT2i, may explain improved cardiac energetics, lowering cardiac afterload and improving ventricular arterial coupling and cardiac efficiency [51] (Figure 3).

Although the precise pathophysiology for SGLT2i antihypertensive effects is not completely understood, they are probably mediated by the osmotic and diuretic effects of SGLT2i because of the inhibition of sodium reabsorption in the kidney proximal tubules. On the other hand, the capability of SGLT2i to promote natriuresis, glucosuria, and osmotic diuresis has been suggested as a mechanism that may ameliorate HF outcomes by favoring hemoconcentration [44]. This theory has been criticized because other diuretic strategies historically failed to impact HF mortality but Hallow et al. proved that dapagliflozin determined a significant reduction in interstitial versus intravascular volume, suggesting that SGLT2i may express a disparity of effect on interstitial and intravascular fluid, so limiting the neurohumoral reflex stimulus that arises in response to intravascular volume reduction with classical diuretics [52].

Interestingly, some evidence suggests that SGLT2i may improve cardiac energetics. Mitochondrial glucose oxidation decreases in the failing heart, determining a reduction in energy production and a fuel-starved heart. The uncoupling between glycolysis and glucose oxidation also leads to increased proton production and, consequently, a decrease in cardiac efficiency (cardiac work/O2 consumed) [53]. Mobilizing adipose tissue fatty acids, SGLT2i increases circulating ketone levels, which maximize cardiac energetics and efficiency, functioning as an economical fuel for the failing heart [54].

A new possible etiology of cardiorenal benefits from SGLT2i therapy is the inhibition of sympathetic nervous system (SNS) activity, indirectly suggested by the observation that SGLT2i reduces blood pressure without increasing heart rate. In fact, emerging data from animal models proved that this class of drugs might lower sympathetic nerve activity, inhibit norepinephrine metabolism in brown adipose tissue, and decrease tyrosine hydroxylase synthesis [55,56].

Adverse remodeling has been highlighted as a crucial determinant in HF pathophysiology. Remarkably, experimental models proved that SGLT2i, i.e., empagliflozin, may reduce left ventricular (LV) mass index when evaluated by cardiac magnetic resonance, compared to placebo [57]; although the molecular pathways of their anti-remodeling effect have not clarified yet, an explanation relies on a decreased fibrosis. On the other hand, in diabetic and non-diabetic rat models, SLGT2i achieved a cardioprotective effect against ischemia/reperfusion injury, determining a reduction in calmodulin kinase II activity, an increased sarcoplasmic reticulum Ca2+ flux and a stronger contractility [58].

These unexpected benefits boosted great interest in SGLT2i, and accumulating evidence have proposed a large range of pathophysiologic effects: inhibition of the cardiac Na+/H+ exchanger, reducing Ca2+ overload [59]; reduction of hyperuricemia [60]; an increase of autophagy and lysosomal degradation, favoring mitochondrial function [61]; a higher erythropoietin secretion [62]; an improved vascular function, by attenuating endothelial cell activation and dysfunction, and those molecular variations accompanying early atherogenesis [63].

#### 3.1.2. SGLT2i and Meta-inflammation

Among all the pathways associated with SGLT2 inhibition, those related to meta-inflammation are probably the most interesting in the pathophysiology of diabetes and HF. Although major cardiovascular outcome trials did not include dosing of inflammatory markers, some insights are available from smaller pilot studies; in fact, SGLT2i therapy has been associated with a decrease in leptin concentration and lower levels of highly sensitive (hs) CRP, tumor necrosis factor-alpha (TNF-α), interleukin-6 (IL-6) and interferon-gamma (IFN-γ) [64].

As mentioned before, adipose tissue may be considered the meta-inflammation fortress. Xu et al. proved that empagliflozin promotes fat utilization and browning of white adipose tissue and reduces M1-polarized macrophage accumulation while promoting the anti-inflammatory M2 macrophage phenotype in white adipose tissue, contributing to a reduction in TNFα plasma levels and mitigation of obesity-related chronic low-grade inflammation [65]. Interestingly, recent theories proposed that leptin synthesis in adipose tissue may determine sodium retention and plasma volume expansion, meta-inflammation, and fibrosis. In this setting, the SGLT2i action on perivisceral adipose tissue may attenuate leptin secretion and, potentially, its paracrine effects, i.e., fibrosis promotion [66].

A similar effect has been described in epicardial fat tissue, considered a paracrine organ able to produce a number of bioactive molecules, i.e., leptin and TNFα, and to influence cardiac remodeling and atherosclerotic plaque in epicardial coronary arteries [67,68]. In particular, in diabetic patients with CAD, SGLT2i decreases epicardial adipose tissue (EAT) and its paracrine products [69].

As regards the potential anti-inflammatory activities of SGLT2 inhibition, animal models proved that empagliflozin therapy mitigates the enhanced expression of inflammatory genes in the diabetic kidney [70]. As proposed in other work from our group, the NLRP3 inflammasome is a milestone in the road that links inflammation and CAD, and its role has also been evaluated in HF patients [71]. Experimental data disclosed that empagliflozin might downregulate the NLRP3 inflammasome, and this result was AMPK-dependent and SGLT2/glucose-lowering independent [72]. As mentioned before, SGLT2i enhances the plasma concentration of ß-hydroxybutyrate, a potent NLRP3 inflammasome blocker, and some authors speculate that beneficial effects of new anti-diabetic drugs may occur secondary to ketone inhibition of the NLRP3 inflammasome [73].

Even if SGLT2i have a role in hyperlipidemia management in diabetic patients, the therapeutic effects on atherosclerosis may be driven by their influence on meta-inflammation. Nasiri-Ansari demonstrated that canagliflozin therapy for 5 weeks significantly reduced atherosclerosis progression in ApoE−/− mice, probably inhibiting *VCAM-1* mRNA expression levels and inflammatory monocyte chemoattractant protein-1 (Mcp-1) expression [74].

Interestingly, SGLT2i may influence atherosclerotic plaque characteristics, favoring stability. In fact, animal models disclosed that treatment with dapagliflozin significantly reduced the growth of cholesterol crystals in atherosclerotic lesions in the diabetic group [75], while canagliflozin increased the collagen content by 1.6-fold in atherosclerotic lesions, leading to plaque stability [74]. Dapagliflozin treatment significantly inhibited cholesterol ester accumulation in macrophages extracted from ApoE−/−mice and reversed the atherosclerosis-associated increase in macrophage infiltration by 20%; finally, it improved the decreased production of smooth muscle cells in animal models [76].

In conclusion, SGLT2i promises to ameliorate the prognosis of patients with and without diabetes with a wide range of molecular pathways, most of them including meta-inflammation and its relationship with HF and atherosclerotic plaque progression.

### 3.2. Glucagon-like Peptide 1 Receptor Agonists (GLP-1 RA)

Glucagon-like peptide-1-(7-36) amide (GLP-1) is a human incretin hormone produced by the gut in response to food. It is primarily an insulin-tropic hormone, playing its role not only to promote insulin secretion but also to inhibit glucagon secretion, regulating the excessive hepatic glucose output and reducing appetite leading to weight loss. Its biological characteristics allow for avoiding hypoglycemia, making GLP-1 an attractive target molecule for anti-diabetic treatment. However, native GLP-1 is rapidly inactivated by the ubiquitous enzyme DPP4; for these reasons, DPP4-resistant GLP-1 analogs have been formulated and approved for medical use in 2005 [77].

Seven randomized control trials (ELIXA-Lixisenatide; LEADER-Liraglutide; SUSTAIN-6—Semaglutide; EXSCEL -Exenatide; HARMONY—Albiglutide; REWIND—Dulaglutide; PIONEER 6—Seraglutide) have explored cardiovascular outcomes in diabetic patients treated with GLP-1 RA [78,79,80,81,82,83,84]. They were large trials of 3000 to 14,000 patients with similar baseline patient demographics and determined to be at high risk or have established cardiovascular disease. All trials were designed to evaluate the primary outcome of MACE and to prove non-inferiority for cardiovascular safety, even if LEADER and HARMONY-OUTCOMES were powered for and demonstrated superiority versus placebo (Table 2).

To sum up, liraglutide, semaglutide, albiglutide, and dulaglutide (but not exenatide or lixisenatide) have proved superior compared to placebo for a reduction in MACE endpoint; however, to date, only liraglutide has shown a decrease in both cardiovascular and all-cause mortality [85]. A recent meta-analysis disclosed that GLP-1 RA reduced all-cause mortality by 12% and adverse renal outcomes by 17%, reporting a significant reduction in hospital admissions for HF, not evidenced by single trials [3].

As far as GLP-1 pathway inhibition is concerned, an emerging drug is drawing attention due to its effects on obesity and weight reduction: Tirzepatide. Tirzepatide is a novel, once-weekly, injectable peptide engineered from the native Glucose-dependent insulinotropic polypeptide (GIP) sequence to have dual GIP/GLP-1-RA activity, which may result in substantial weight reduction, even more than GLP-1-RA alone, by addressing several pathways involved in energy homeostasis. Tirzepatide has been approved by the FDA for the treatment of T2DM with safety and tolerability characteristics similar to other incretin-based therapies. Despite significant improvements in cardiometabolic measures have been observed, no information is available on long-term CV effects. Further studies may evaluate possible unexplored mechanisms linking these new anti-diabetic drugs to CAD and meta-inflammation [86].
ijms-24-08643-t002_Table 2Table 2Cardiovascular Outcome Trial and GLP-1RA.GLP-1RA NameClinical Trial Name (CVOT)Brief ReferenceNumber of PatientsDefinition of CVOutcomesMedian Follow Up (Months)Principal FindingsLixisenatideELIXAHolman RR et al., New England Journal of Medicine, 2017 [78]6068CV death, myocardial infarction, stroke, or hospitalization for unstable angina25The primary endpoint (a composite of the first occurrence of any of the following: death from CV causes, non-fatal myocardial infarction, non-fatal stroke, or hospitalization for unstable angina) event occurred in 13.4% of the lixisenatide group and in 13.2% of the placebo group (HR, 1.02; 95% CI, 0.89 to 1.17), which showed the non-inferiority of lixisenatide to placebo (*p* < 0.001) but did not show superiority (*p* = 0.81).LiraglutideLEADERMarso SP et al.,New England Journal of Medicine, 2016 [79]9340The first occurrence of death from CV causes non-fatal myocardial infarction or non-fatal stroke15.8The primary outcome (the first occurrence of death from CV causes, non-fatal myocardial infarction, or non-fatal stroke) occurred in significantly fewer patients in the liraglutide group (13.0%) than in the placebo group (14.9%) (HR, 0.87; 95% CI, 0.78 to 0.97; *p* < 0.001 for non-inferiority; *p* = 0.01 for superiority). Fewer patients died from cardiovascular causes in the liraglutide group (4.7%) than in the placebo group (6.0%) (HR, 0.78; 95% CI, 0.66 to 0.93; *p* = 0.007).Mann JFE et al.,Circulation, 2018 [87]9340Composite of CVdeath, non-fatal myocardial infarction, or non-fatal stroke36In patients with eGFR < 60 mL/min/1.73 m^2^, risk reduction for the primary composite CV outcome with liraglutide was greater (HR, 0.69; 95% CI, 0.57–0.85) versus those with eGFR ≥ 60 mL/min/1.73 m^2^ (HR, 0.94; 95% CI, 0.83–1.07; interaction *p* = 0.01).SemaglutideSUSTAIN-6Marso SP et al.,New England Journal of Medicine, 2016 [80]3297The first occurrence of CV death, non-fatal myocardial infarction, or non-fatal stroke24The primary outcome (composite of first occurrence of death from CV causes, non-fatal myocardial infarction, or non-fatal stroke. Occurred in 6.6% of the semaglutide group and in 8.9% of the placebo group (HR, 0.74; 95% CI, 0.58 to 0.95; *p* < 0.001 for non-inferiority). Non-fatal myocardial infarction occurred in 2.9% of the patients receiving semaglutide and in 3.9% of those receiving placebo (HR, 0.74; 95% CI, 0.51 to 1.08; *p* = 0.12); non-fatal stroke occurred in 1.6% and 2.7%, respectively (HR, 0.61; 95% CI, 0.38 to 0.99; *p* = 0.04).
PIONEER-6Husain M et al.,New England Journal of Medicine, 2019 [84]3183The first occurrence of CV death, non-fatal myocardial infarction, or non-fatal stroke15.9Major adverse CV events occurred in 3.8% of the oral semaglutide group and in 4.8% of the placebo group (hazard ratio, 0.79; 95% CI, 0.57 to 1.11; *p* < 0.001 for non-inferiority).ExenatideEXSCELHolman RR et al.,New England Journal of Medicine, 2016 [78]14,752The first occurrence of death from CV causes non-fatal myocardial infarction or non-fatal stroke38.4A primary outcome (composite of the first occurrence of any component of the composite outcome of death from CV causes, non-fatal myocardial infarction, or non-fatal stroke) event occurred in 11.4% in the exenatide group and in 12.2% in the placebo group (HR, 0.91; 95% CI, 0.83 to 1.00)DulaglutideREWINDGerstein HC et al.,Lancet, 2019 [83]9901The first occurrence of the composite endpoint of non-fatal myocardial infarction, non-fatal stroke, or CV death64.8The primary outcome (composite of first occurrence of non-fatal myocardial infarction, non-fatal stroke, and death from CV causes or unknown causes) occurred in 12.0% in the dulaglutide group and in 13.4% in the placebo group (HR 0.88, 95% CI 0.79–0.99; *p* = 0.026).Confidence interval, CI; CVOT, Cardiovascular Outcome Trial; CV, cardiovascular; Hazard ratio, HR.


#### Possible Target of GLP-1 RA Pharmacodynamics and Their Link with Meta-Inflammation

As seen before for SGLT2i, the cardiovascular outcomes revealed in large trials are likely to be independent of glycemic control since just a modest reduction in glycosylated hemoglobin levels has been reached in the treatment groups.

As mentioned for SGLT2i, reduction in blood pressure, modification of heart rate, and weight loss have been highlighted in cardiovascular outcome trials, but their variations are probably too small to justify the significant benefits observed in clinical studies [88]. Likewise, GLP-1 RA may affect lipidic profiles in both normoglycemic and diabetic rodents and humans; in fact, they proved to rapidly decrease intestinal chylomicron production and secretion, independently of differences in gastric emptying [89]. On the other hand, preclinical data sustained meta-inflammation links antiatherogenic and plaque stabilization effects of GLP-1 RA as the leading hypothesis for the beneficial cardiovascular effects of this class of drugs (Figure 3).

Multiple evidence proved that GLP-1 secretion from enteroendocrine cells is alternatively stimulated or inhibited by pro-inflammatory stimuli describing an intricated pathway in which GLP-1 is both a target and a mediator of the inflammatory response. In some mice models, Interleukin-1 (IL-1), IL-6, and LPS all acutely increased plasma levels of GLP-1 [90], while the demonstration that *Glp1r*−/− mice exhibit gut microbial dysbiosis and markedly increased sensitivity to experimental colonic inflammation proved the importance of intestinal GLP-1 signaling for control of local inflammatory signals [91]. In other preclinical studies, GLP-1 reduced vascular monocyte adhesion and macrophage accumulation in blood vessels both in normoglycemic or hyperglycemic mice at high risk for the development of experimental atherosclerosis, along with other anti-inflammatory effects, i.e., reduced plaque macrophage and matrix metalloproteinase-9 (MMP-9) accumulation [92] (Figure 4). In human studies, exenatide (10 mg twice daily for 12 weeks) in obese and diabetic patients reduced circulating levels of MCP-1, serum amyloid A, MMP-9, and IL-6 [93]. Due to its complementary mechanism of action on insulin, glucagon, and appetite, GLP-1 RA therapy has been associated with lower glucose variability, i.e., fluctuations of glucose levels, either within or between days, which plays an important role in favoring the appearance of endothelial dysfunction, oxidative stress, and inflammation [94]. In this setting, native GLP-1 infusion over several hours recovers endothelial function; in particular, GLP-1 degradation products, i.e., GLP-1(9-36) or GLP-1(28-36), somewhat account for the vasodilatory actions ascribed to native GLP-1 [95]. Furthermore, therapy with twice-daily exenatide ameliorated postprandial endothelial function, assessed by measurement of digital hyperemia in diabetic patients [96]. Interestingly, preclinical data disclosed that GLP-1RA might directly inhibit mouse and human platelet aggregation and thrombus formation ex vivo [97], even if adult human platelets express a functional canonical GLP-1 receptor remains uncertain.

In the last years, GLP-1 RA raised great interest because of the hypothesis of cardioprotection in the setting of ischemia-reperfusion damage. In fact, preclinical studies revealed significant cardioprotection after the administration of GLP-1RA in the setting of permanent or transient left anterior descending coronary artery occlusion: GLP-1RA treatment was associated with reduced infarct size, improved survival, and preservation of ventricular function [98]. Additionally, in a randomized, placebo-controlled study, exenatide for 3 days in patients with ST-segment elevation myocardial infarction (STEMI) treated with percutaneous angioplasty, reduced levels of creatine kinase-MB and troponin I and ameliorated infarct size, assessed by cardiac MRI 38 days after reperfusion [99]. Furthermore, improvement in diastolic function and global longitudinal strain after transient exenatide administration have been demonstrated at the echocardiographic analysis at 6 months follow-up.

Literature has shown that EAT, a milestone of meta-inflammation, may be a modifiable and measurable cardiovascular risk factor, and that is thicker in subjects with DM, probably because of its solid relationship with visceral adiposity and insulin resistance [100]. Its anatomical position and pro-inflammatory transcriptome pave the way to the development and progression of atherosclerosis and CAD. Interestingly, Iacobellis et al. demonstrated that weekly administration of either GLP-1 RA semaglutide or dulaglutide produces a rapid, substantial, and dose-dependent reduction in EAT thickness [101], and although this action is shared with SGLT1, it is stronger [102].

In conclusion, the cardiovascular benefits of GLP-1 may be dependent on the complex relationship of this incretin hormone with the atherosclerotic pathways, in particular those having meta-inflammation and plaque stability as their intermediary.

## 4. From the Bed to the Bench Side: The New Glucose-Lowering Drugs and Related Cardiovascular Benefits

New anti-diabetic drugs have proven unexpected cardiovascular outcomes, which promise to significantly modify the management of this class of patients. Nevertheless, it remains doubts on the exact mechanisms through which SGLT2i and GLP-1 RA reach their exciting clinical benefits in patients with HF and cardiac ischemic disease. Clinical results indicate that these data may hardly be fully explained by glucose lowering and that the putative mechanisms may act on top of excellent background therapy, may have a rapid onset of the benefit, and may include renal protection, especially as far as SGLT2i are concerned [73].

As mentioned before, new glucose-lowering drugs proved to substantially affect inflammation pathways, in particular meta-inflammation, the low-grade chronic and sterile inflammatory status maintained by high nutrient intake, which plays a crucial role in the pathophysiology of DM and CAD. SGLT-2i have demonstrated to downregulate inflammatory cytokines; suppress cholesterol esters accumulation; decrease fat cell size; improve macrophage polarization and limit their transformation to foam cells; reduce intra-plaque macrophage infiltration; restore the reduced content of smooth muscular cells and increase collagen content in atherosclerotic lesions. Beyond the notorious effect on HF, SGLT-2i proved to slow atherosclerosis progression, favor the stability of already-formed atherosclerotic lesions, and, eventually, improve the cardiovascular-related prognosis of patients with diabetes (Figure 3). On the other hand, GLP-1 RA showed similar cardiovascular outcomes, in particular in patients affected by diabetes and CAD, acting on lipid profile, endothelial cell function, ischemia-reperfusion processes, and inflammation pathways, such as the downregulation of multiples cytokines and the restraint on local intestinal inflammation.

The never-ending “love” story between inflammation and cardiovascular diseases has its roots in the last decade of the former century [103]. Since then, growing data have supported the concept of inflammation as a milestone in heart disease pathophysiology. Recently, great emphasis has been reached because of data from the Canakinumab Anti-inflammatory Thrombosis Outcome Study (CANTOS), which supported that reducing inflammation by targeting IL-1β innate immunity pathway with canakinumab directly confers the cardiovascular benefit in a high-risk population [68]. In this framework, the connection between SGLT-2i, GLP-1 RA, and meta-inflammation rails the same path, reinforcing the strength of this relation.

## 5. Conclusions

A growing body of emerging data suggests that new anti-diabetic drugs may control low-grade inflammation and reduce mRNA expression of some cytokines and chemokines, significantly modifying the molecular milieu in which classical determinants of CAD act and limiting the boost imposed by meta-inflammation. For these reasons, increasing emphasis has now been placed on immunomodulatory treatments and agents that can reduce low-grade inflammation in the management of diabetes: targeting inflammation in diabetic patients may be a complementary therapeutic possibility beyond strict glycemic control.

The molecular mechanisms underpinning these results are probably multifaceted and cannot be explained by the only link with meta-inflammation. However, a decrease in adipose tissue inflammation, a switch towards a less inflammatory environment, and the attenuation of postprandial hyperglycemia, along with an improvement in renal and general hemodynamics, may play a role in the reduction of cardiovascular mortality and morbidity. Further biological and clinical studies are needed to improve our knowledge of the molecular mechanisms and off-target effects, which link meta-inflammation and diabetes to finally knock down the residual cardiovascular risk and improve the lifetime benefits in HF patients.

## Figures and Tables

**Figure 1 ijms-24-08643-f001:**
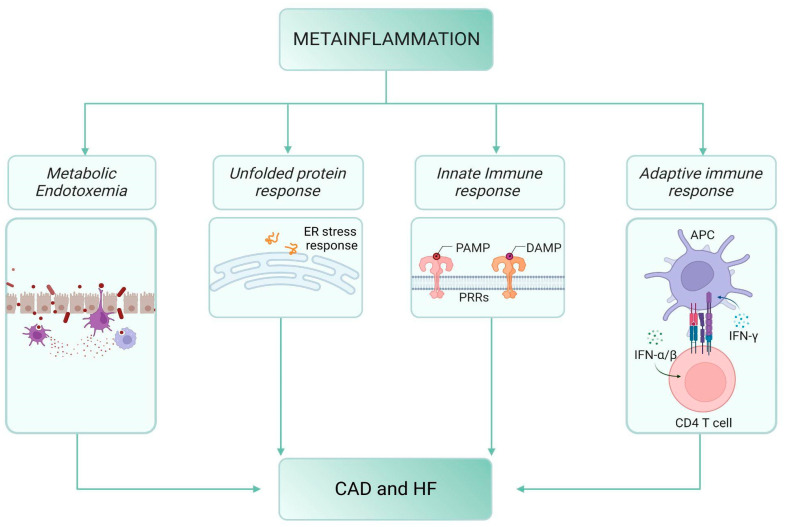
Mechanisms linking meta-inflammation, diabetes, and cardiovascular disease. Meta-inflammation may induce CAD onset and progression and HF through a low-grade-inflammatory state. The mechanisms involved in the onset of this metabolic-–inflammatory status start from metabolic endotoxemia, with the increase of LPS, leukocyte activation, and local and systemic inflammation. The unfolded protein response also enhances pro-inflammatory action. Finally, both innate and immune responses arouse complex pathways setting the framework for the inflammatory status. CAD, cardiovascular diseases. HF, heart failure.

**Figure 2 ijms-24-08643-f002:**
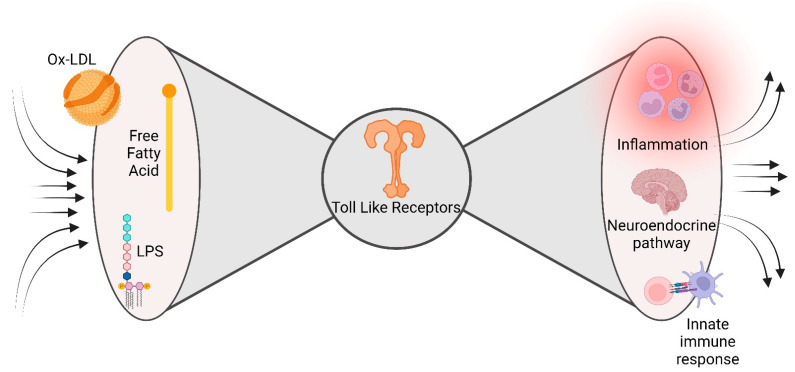
Bow tie model. The characteristic of bow tie architecture is the ability to draw in a wide variety of inputs, such as self- and non-self-stimuli (free fatty acids, LPS, and oxidized-LDL) into the core of components (toll-like receptors), which can convert the inputs into a wide range of outputs, such as a variety of inflammatory components.

**Figure 3 ijms-24-08643-f003:**
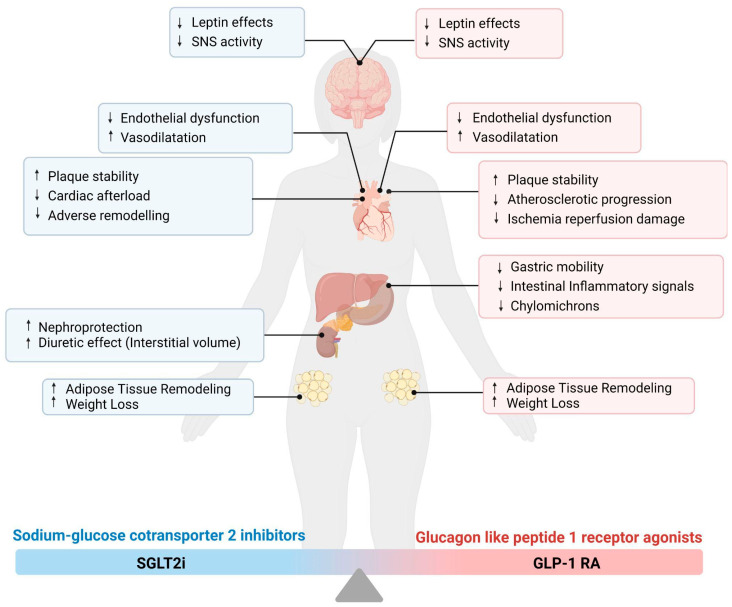
Possible effects of SGLT2i and GLP-1 RA pharmacodynamics. Both drugs act similarly on the SNS, weight loss and endothelial dysfunction, and vasodilatation mechanisms. On the other hand, major differences emerge in the effect on the cardiovascular system. Indeed, SGLT2i lower cardiac afterload and adverse remodeling, while GLP-1 RA contributes to mitigating atherosclerotic progression and ischemia-reperfusion damage. Up arrow stands for an increase in the reported action. Down arrow stands for a decrease in the reported action. SGLT2i, Sodium–glucose cotransporter 2 inhibitors; GLP-1 RA, Glucagon-Like Peptide 1 Receptor Agonists; SNS, sympathetic nervous system.

**Figure 4 ijms-24-08643-f004:**
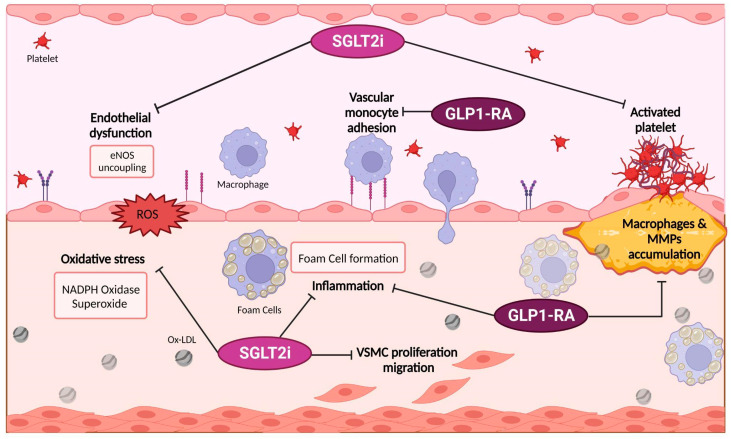
Molecular mechanisms of new glucose-lowering drugs. SGLT2i proved to reduce endothelial dysfunction and oxidative stress. Furthermore, SGLT2i may decrease platelet activation and vascular smooth muscle cell proliferation and migration. On the other hand, GLP1-RA significantly limits vascular monocyte adhesion and macrophages and metalloproteinases accumulation in the atherosclerotic plaque. Remarkably, both drugs showed a reduction in inflammatory response. SGLT2i, Sodium–glucose cotransporter 2 inhibitors; GLP-1 RA, Glucagon-Like Peptide 1 Receptor Agonists; eNOS, endothelial nitric oxide synthase; NADPH, nicotinamide adenine dinucleotide phosphate; MMPs, metalloprotease; Ox-LDL, oxidized LDL; VSMC, vascular smooth muscle cells.

**Table 1 ijms-24-08643-t001:** Cardiovascular Outcome Trial and SGLT-2i.

SGLT-2i Name	Clinical Trial Name (CVOT)	Brief Reference	Number of Patients	Definition of CV Outcomes	Median Follow Up (Months)	Principal Findings
Empagliflozin	EMPA-REG OUTCOME	Zinman B et al., New England Journal of Medicine, 2015 [44]	7020	CV, non-fatal myocardial infarction, or non-fatal stroke	37.2	The empagliflozin group presented significantly lower rates of death from CV causes (3.7% vs. 5.9% in the placebo group; 38% relative risk reduction), and hospitalization for HF (2.7% and 4.1%, respectively; 35% relative risk reduction).
Canagliflozin	CANVAS	Neal B et al., New England Journal of Medicine, 2017 [45]	10,142	CV death or hospitalization for HF	29	The rate of the primary outcome (composite of death from cardiovascular causes, non-fatal myocardial infarction, or non-fatal stroke) was lower with canagliflozin than with placebo (HR, 0.86; 95% CI, 0.75 to 0.97; *p* < 0.001 for non-inferiority; *p* = 0.02 for superiority).
Dapagliflozin	DECLARE-TIMI	Wiviott SD et al., New England Journal of Medicine, 2019 [47]	17,160	CV death or hospitalization for HF	50.4	Dapagliflozin did result in a lower rate of CV death or hospitalization for heart failure (4.9% vs. 5.8%; HR, 0.83; 95% CI, 0.73 to 0.95; *p* = 0.005), which reflected a lower rate of hospitalization for HF (hazard ratio, 0.73; 95% CI, 0.61 to 0.88)
Dapagliflozin	DAPA-HF	McMurray JJV et al., New England Journal of Medicine, 2019 [48]	4744	CV death or HF hospitalization/urgent HF visit	18.2	The primary outcome (composite of worsening heart failure or death from cardiovascular causes) was lower in the dapagliflozin group than in the placebo group (HR, 0.74; 95% CI, 0.65 to 0.85; *p* < 0.001). The first worsening HF event occurred in 10.0% of the dapagliflozin group and in 13.7% in the placebo group (HR, 0.70; 95% CI, 0.59 to 0.83). Death from cardiovascular causes occurred in 9.6% in the dapagliflozin group and in 11.5% in the placebo group (HR, 0.82; 95% CI, 0.69 to 0.98)
Ertugliflozin	VERTIS-CV	Cannon et al., New England Journal of Medicine, 2020 [49]	8246	CV death or hospitalization for HF	42	MACE occurred in 11.9% of the ertugliflozin group and in 11.9% of the placebo group (HR, 0.97; 95.6% CI, 0.85 to 1.11; *p* < 0.001 for non-inferiority). Death from CV causes or hospitalization for HF occurred in 8.1% in the ertugliflozin group and in 9.1% in the placebo group (HR, 0.88; 95.8% CI, 0.75 to 1.03; *p* = 0.11 for superiority).
Sotagliflozin	SCORED	Bahtt DL et al., New England Journal of Medicine, 2021 [50]	10,584	CV death or HF hospitalization/ urgent HF visit	16	The rate of primary end-point events was 5.6 events in the sotagliflozin group and 7.5 events per 100 patient-years in the placebo group (HR, 0.74; 95% [CI], 0.63 to 0.88; *p* < 0.001). The rate of deaths from cardiovascular causes per 100 patient-years was 2.2 with sotagliflozin and 2.4 with placebo (HR, 0.90; 95% CI, 0.73 to 1.12; *p* = 0.35).

Confidence interval, CI; CVOT, Cardiovascular Outcome Trial; CV, cardiovascular; Hazard ratio, HR; HF, heart failure.

## Data Availability

Not applicable.

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
