# Peer review of "Meta-Inflammation and New Anti-Diabetic Drugs: A New Chance to Knock Down Residual Cardiovascular Risk"

_ijms, 2023, doi:10.3390/ijms24108643_

Round 1

Reviewer 1 Report

In several words of the text, correct when using letters of the Greek alphabet, as an example beta cells of the pancreas, NF Kapa beta, among others, line 65

In my opinion, the text should be reviewed for the English language used, only as proof

Author Response

In several words of the text, correct when using letters of the Greek alphabet, as an example beta cells of the pancreas, NF Kapa beta, among others, line 65.

We revised the manuscript as indicated by the reviewer.

Reviewer 2 Report

In this article, d’Aiello et al. review the effect of anti-diabetic drugs to reduce meta-inflammation as cardiovascular disease (CVD) complications and risks. The article comprehensively discusses the major factors, immune cell types, and pathways involved in meta-inflammation. They also discuss different anti-diabetic therapeutics (SGLT2i and GLP1-RA) and their impact on CVD risk. The diagrams are well-made. However, the authors need to address these concerns before the acceptance of the article:

  1. Although the article covers important topics comprehensively, the major caveat of this study is the writing style and grammar. Because of the minor grammatical errors throughout the article, including sentence structure, tenses, spelling, unnecessary spaces, etc., it takes time to understand what the authors want to convey. A thorough proofreading of the entire article by a native English speaker is warranted. To provide a single example: the authors in one section refer to lower glucose drugs instead of glucose-lowering drugs. Such mistakes should also be addressed throughout the article.
  2. The abstract is short. A line or two about what the article will cover is necessary within the abstract.
  3. A section on diabetes and CVD is essential in the earlier part of the article before delving into how anti-diabetic drugs are protective against CVD complications.
  4. Figures: The authors need to add detailed figure legends describing the figures. Also, the font sizes of the text on the figures should be increased.
  1. The article has minor grammatical errors, including spaces, spelling, and punctuation. Please perform thorough proofreading of the article.

Reviewer 3 Report

In Metainflammation and new anti-diabetic drugs: a new chance to knock down residual cardiovascular risk, Alessia d’Aiello et al analyzed the impact of sodium-glucose cotransporter 2 inhibitors (SGLT2i) and glucagon like peptide 1 receptor agonists (GLP-1 RAs) on reduction of incidence of major adverse cardiovascular events (MACEs). The analysis was focused on the connection between diabetes and metainflammation and the potential regulation of SGLT2i and GLP-1 RAs on the reduction of cardiovascular risk. However, there are several points:

  1. It is recommendable to include new figures that analyze the representative cellular effects of SGLT2i and GLP-1 RAs.
  2. Information in lanes 126-129 is confusing.
  3. The analysis of the “bow tie” architecture is not explicit.
  4. Information in Table 1 could be broadened.
  5. The manuscript is focused on the impact of metainflammation, and the authors analyze in depth the association of SGLTi and metainflammation; however, this aspect was not addressed for GLP-1 RAs. Moreover, the impact on critical inflammatory targets such as inflammasome was not analyzed in detail.
  6. A special case would be the analysis of tirzepatide.

Round 2

Reviewer 2 Report

The article is significantly improved. Accept it in the current form.

Reviewer 3 Report

Dear authors

I suggest the approval of the manuscript.

Best regards,